# Discovering Symbolic Models from Deep Learning with Inductive Biases

**Miles Cranmer**[1]  **Alvaro Sanchez-Gonzalez**[2]  **Peter Battaglia**[2]  **Rui Xu**[1]

**Kyle Cranmer**[3]  **David Spergel**[4,1]  **Shirley Ho**[4,3,1,5]

[1] Princeton University, Princeton, USA  [2] DeepMind, London, UK
[3] New York University, New York City, USA  [4] Flatiron Institute, New York City, USA
[5] Carnegie Mellon University, Pittsburgh, USA

## Abstract

We develop a general approach to distill symbolic representations of a learned deep model by introducing strong inductive biases. We focus on Graph Neural Networks (GNNs). The technique works as follows: we first encourage sparse latent representations when we train a GNN in a supervised setting, then we apply symbolic regression to components of the learned model to extract explicit physical relations. We find the correct known equations, including force laws and Hamiltonians, can be extracted from the neural network. We then apply our method to a non-trivial cosmology example—a detailed dark matter simulation—and discover a new analytic formula which can predict the concentration of dark matter from the mass distribution of nearby cosmic structures. The symbolic expressions extracted from the GNN using our technique also generalized to out-of-distribution-data better than the GNN itself. Our approach offers alternative directions for interpreting neural networks and discovering novel physical principles from the representations they learn.

## 1  Introduction

*The miracle of the appropriateness of the language of mathematics for the formulation of the laws of physics is a wonderful gift which we neither understand nor deserve. We should be grateful for it and hope that it will remain valid in future research and that it will extend, for better or for worse, to our pleasure, even though perhaps also to our bafflement, to wide branches of learning.*—Eugene Wigner "The Unreasonable Effectiveness of Mathematics in the Natural Sciences" [1].

For thousands of years, science has leveraged models made out of closed-form symbolic expressions, thanks to their many advantages: algebraic expressions are usually compact, present explicit interpretations, and generalize well. However, finding these algebraic expressions is difficult. Symbolic regression is one option: a supervised machine learning technique that assembles analytic functions to model a given dataset. However, typically one uses genetic algorithms—essentially a brute force procedure as in [2]—which scale exponentially with the number of input variables and operators. Many machine learning problems are thus intractable for traditional symbolic regression.

On the other hand, deep learning methods allow efficient training of complex models on high-dimensional datasets. However, these learned models are black boxes, and difficult to interpret.

---

Code for our models and experiments can be found at `https://github.com/MilesCranmer/symbolic_deep_learning`.

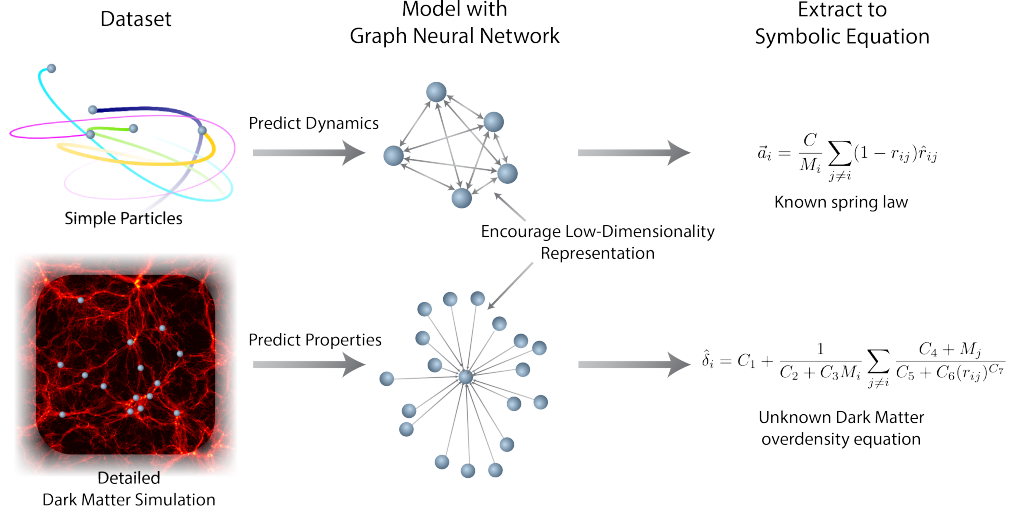

Figure 1: A cartoon depicting how we extract physical equations from a dataset.

Furthermore, generalization is difficult without prior knowledge about the data imposed directly on the model. Even if we impose strong inductive biases on the models to improve generalization, the learned parts of networks typically are linear piece-wise approximations which extrapolate linearly (if using ReLU as activation [3]).

Here, we propose a general framework to leverage the advantages of both deep learning and symbolic regression. As an example, we study Graph Networks (GNs or GNNs) [4] as they have strong and well-motivated inductive biases that are very well suited to problems we are interested in. Then we apply symbolic regression to fit different internal parts of the learned model that operate on reduced size representations. The symbolic expressions can then be joined together, giving rise to an overall algebraic equation equivalent to the trained GN. Our work is a generalized and extended version of that in [5].

We apply our framework to three problems—rediscovering force laws, rediscovering Hamiltonians, and a real world astrophysical challenge—and demonstrate that we can drastically improve generalization, and distill plausible analytical expressions. We not only recover the injected closed-form physical laws for Newtonian and Hamiltonian examples, but we also derive a new interpretable closed-form analytical expression that can be useful in astrophysics.

## 2 Framework

Our framework can be summarized as follows. (1) Engineer a deep learning model with a separable internal structure that provides an inductive bias well matched to the nature of the data. Specifically, in the case of interacting particles, we use Graph Networks as the core inductive bias in our models. (2) Train the model end-to-end using available data. (3) Fit symbolic expressions to the distinct functions learned by the model internally. (4) Replace these functions in the deep model by the symbolic expressions. This procedure with the potential to discover new symbolic expressions for non-trivial datasets is illustrated in fig. 1.

**Particle systems and Graph Networks.** In this paper we focus on problems that can be well described as interacting particle systems. Nearly all of the physics we experience in our day-to-day life can be described in terms of interactions rules between particles or entities, so this is broadly relevant. Recent work has leveraged the inductive biases of Interaction Networks (INs) [6] in their generalized form, the *Graph Network*, a type of Graph Neural Network [7, 8, 9], to learn models of particle systems in many physical domains [6, 10, 11, 12, 13, 14, 15, 16].

Therefore we use Graph Networks (GNs) at the core of our models, and incorporate into them physically motivated inductive biases appropriate for each of our case studies. Some other interesting

approaches for learning low-dimensional general dynamical models include [17, 18, 19]. Other related work which studies the physical reasoning abilities of deep models include [20, 21, 22].

Internally, GNs are structured into three distinct components: an edge model, a node model, and a global model, which take on different but explicit roles in a regression problem. The edge model, or "message function," which we denote by $\phi^e$, maps from a pair of nodes ($\mathbf{v}_i, \mathbf{v}_j \in \mathbb{R}^{L_v}$) connected by an edge in a graph together with some vector information for the edge, to a message vector. These message vectors are summed element-wise for each receiving node over all of their sending nodes, and the summed vector is passed to the node model. The node model, $\phi^v$, takes the receiving node and the summed message vector, and computes an updated node: a vector representing some property or dynamical update. Finally, a global model $\phi^u$ aggregates all messages and all updated nodes and computes a global property. $\phi^e$, $\phi^v$, $\phi^u$ are usually approximated using multilayer-perceptrons, making them differentiable end-to-end. More details on GNs are given in the appendix. We illustrate the internal structure of a GN in fig. 2.

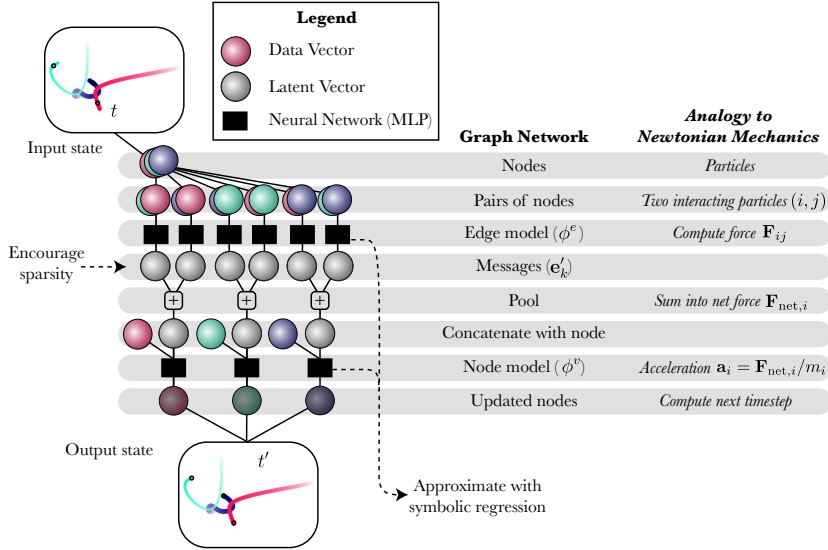

Figure 2: An illustration of the internal structure of the graph neural network we use in some of our experiments. Note that the comparison to Newtonian mechanics is purely for explanatory purposes, but is not explicit. Differences include: the "forces" (messages) are often high dimensional, the nodes need not be physical particles, $\phi^e$ and $\phi^v$ are arbitrary learned functions, and the output need not be an updated state. However, the rough equivalency between this architecture and physical frameworks allows us to interpret learned formulas in terms of existing physics.

GNs are the ideal candidate for our approach due to their inductive biases shared by many physics problems. (a) They are equivariant under particle permutations. (b) They are differentiable end-to-end and can be trained efficiently using gradient descent. (c) They make use of three separate and interpretable internal functions $\phi^e$, $\phi^v$, $\phi^u$, which are our targets for the symbolic regression. GNs can also be embedded with additional symmetries as in [23, 24], but we do not implement these.

**Symbolic regression.**   After training the Graph Networks, we use the symbolic regression package *eureqa* [2] to perform symbolic regression and fit compact closed-form analytical expressions to $\phi^e$, $\phi^v$, and $\phi^u$ independently. *eureqa* works by using a genetic algorithm to combine algebraic expressions stochastically. The technique is analogous to natural selection, where the "fitness" of each expression is defined in terms of simplicity and accuracy. The operators considered in the fitting process are $+, -, \times, /, >, <, \wedge, \exp, \log, \mathrm{IF}(\cdot, \cdot, \cdot)$ as well as real constants. After fitting expressions to each part of the graph network, we substitute the expressions into the model to create an alternative analytic model. We then refit any parameters in the symbolic model to the data a second time, to avoid the accumulated approximation error. Further details are given in the appendix.

This approach gives us an explicit way of interpreting the trained weights of a structured neural network. It also allows us to extend symbolic regression to high-dimensional datasets, where it

is otherwise intractable. As an example, consider attempting to discover the relationship between a scalar and a time series, given data $\{(z_i, \{\mathbf{x}_{i,1}, \mathbf{x}_{i,2}, \ldots \mathbf{x}_{i,100}\})\}$, where $z_i \in \mathbb{R}$ and $\mathbf{x}_{i,j} \in \mathbb{R}^5$. Assume the true relationship as $z_i = y_i^2$, for $y_i = \sum_{j=1}^{100} y_{i,j}$, $y_{i,j} = \exp(x_{i,j,3}) + \cos(2x_{i,j,1})$. Now, in a learnable model, assume an inductive bias $z_i = f(\sum_{j=1}^{100} g(\mathbf{x}_{i,j}))$ for scalar functions $f$ and $g$. If we need to consider $10^9$ equations for both $f$ and $g$, then a standard symbolic regression search would need to consider their combination, leading to $(10^9)^2 = 10^{18}$ equations in total. But if we first fit a neural network for $f$ and $g$, and after training, fit an equation to $f$ and $g$ *separately*, we only need to consider $2 \times 10^9$ equations. In effect, we factorize high-dimensional datasets into smaller sub-problems that are tractable for symbolic regression.

We emphasize that this method is not a new symbolic regression technique by itself; rather, it is a way of extending any existing symbolic regression method to high-dimensional datasets by the use of a neural network with a well-motivated inductive bias. While we chose *eureqa* for our experiments based on its efficiency and ease-of-use, we could have chosen another low-dimensional symbolic regression package, such as our new high-performance package *PySR*[1] [25]. Other community packages such as [26, 27, 28, 29, 30, 31, 32, 33, 34], could likely also be used and achieve similar results (although [30] appears to not fit constants in equations, so could not be used here). Ref. [27] is an interesting approach that uses gradient descent on a pre-defined equation up to some depth, parametrized with a neural network, instead of genetic algorithms; [33] uses gradient descent on a latent embedding of an equation; and [34] demonstrates Monte Carlo Tree Search as a symbolic regression technique, using an asymptotic constraint as input to a neural network which guides the search. These could all be used as drop-in replacements for *eureqa* here to extend their algorithms to high-dimensional datasets.

We also note several exciting packages for symbolic regression of partial differential equations on gridded data: [35, 36, 37, 38, 39, 40]. These either use sparse regression of coefficients over a library of PDE operators, or a genetic algorithm. While not applicable to our use-cases, these would be interesting to consider for future extensions to gridded PDE data.

**Compact internal representations.** While training, we encourage the model to use compact internal representations for latent hidden features (e.g., messages) by adding regularization terms to the loss (we investigate using $L_1$ and KL penalty terms with a fixed prior, see more details in the Appendix). One motivation for doing this is based on *Occam's Razor*: science always prefers the simpler model or representation of two which give similar accuracy. Another stronger motivation is that if there is a law that perfectly describes a system in terms of summed message vectors in a compact space (what we call a linear latent space), then we expect that a trained GN, with message vectors of the same dimension as that latent space, will be mathematical rotations of the true vectors. We give a mathematical explanation of this reasoning in the appendix, and emphasize that while it may seem obvious now, our work is the first to demonstrate it. More practically, by reducing the size of the latent representations, we can filter out all low-variance latent features without compromising the accuracy of the model, and vastly reducing the dimensionality of the hidden vectors. This makes the symbolic regression of the internal models more tractable.

**Implementation details.** We write our models with PyTorch [41] and PyTorch Geometric[42]. We train them with a decaying learning schedule using Adam [43]. The symbolic regression technique is described in section 4.1. More details are provided in the Appendix.

## 3 Case studies

In this section we present three specific case studies where we apply our proposed framework using additional inductive biases.

**Newtonian dynamics.** Newtonian dynamics describes the dynamics of particles according to Newton's law of motion: the motion of each particle is modeled using incident forces from nearby particles, which change its position, velocity and acceleration. Many important forces in physics (e.g., gravitational force $-\frac{Gm_1m_2}{r^2}\hat{r}$) are defined on pairs of particles, analogous to the message

function $\phi^e$ of our Graph Networks. The summation that aggregates messages is analogous to the calculation of the net force on a receiving particle. Finally, the node function, $\phi^v$, acts like Newton's law: acceleration equals the net force (the summed message) divided by the mass of the receiving particle.

To train a model on Newtonian dynamics data, we train the GN to predict the instantaneous acceleration of the particle against that calculated in the simulation. While Newtonian mechanics inspired the original development of INs, never before has an attempt to distill the relationship between the forces and the learned messages been successful. When applying the framework to this Newtonian dynamics problem (as illustrated in fig. 1), we expect the model trained with our framework to discover that the optimal dimensionality of messages should match the number of spatial dimensions. We also expect to recover algebraic formulas for pairwise interactions, and generalize better than purely learned models. We refer our readers to section 4.1 and the Appendix for more details.

**Hamiltonian dynamics.** Hamiltonian dynamics describes a system's total energy $\mathcal{H}(\mathbf{q}, \mathbf{p})$ as a function of its canonical coordinates $\mathbf{q}$ and momenta $\mathbf{p}$—e.g., each particle's position and momentum. The dynamics of the system change perpendicularly to the gradient of $\mathcal{H}$: $\frac{d\mathbf{q}}{dt} = \frac{\partial \mathcal{H}}{\partial \mathbf{p}}, \frac{d\mathbf{p}}{dt} = -\frac{d\mathcal{H}}{d\mathbf{q}}$.

Here, we will use a variant of a Hamiltonian Graph Network (HGN) [44] to learn $\mathcal{H}$ for the Newtonian dynamics data. This model is a combination of a Hamiltonian Neural Network [45, 46] and GN. In this case, the global model $\phi^u$ of the GN will output a single scalar value for the entire system representing the energy, and hence the GN will have the same functional form as a Hamiltonian. By then taking the partial derivatives of the GN-predicted $\mathcal{H}$ with respect to the position and momentum, $\mathbf{q}$ and $\mathbf{p}$, respectively, of the input nodes, we will be able to calculate the updates to the momentum and position. We impose a modification to the HGN to facilitate its interpretability, and name this the "Flattened HGN" or FlatHGN: instead of summing high-dimensional encodings of nodes to calculate $\phi^u$, we instead set it to be a sum of scalar pairwise interaction terms, $\mathcal{H}_{\text{pair}}$ and a per-particle term, $\mathcal{H}_{\text{self}}$. This is because many physical systems can be exactly described this way. This is a Hamiltonian version of the Lagrangian Graph Network in [47], and is similar to [48]. This is still general enough to express many physical systems, as nearly all of physics can be written as summed interaction energies, but could also be relaxed in the context of the framework.

Even though the model is trained end-to-end, we expect our framework to allow us to extract analytical expressions for both the per-particle kinetic energy, and the scalar pairwise potential energy. We refer our readers to our section 4.2 and the Appendix for more details.

**Dark matter halos for cosmology.** We also apply our framework to a dataset generated from state-of-the-art dark matter simulations [49]. We predict a property ("overdensity") of a dark matter blob (called a "halo") from the properties (positions, velocities, masses) of halos nearby. We would like to extract this relationship as an analytic expression so we may interpret it theoretically. This problem differs from the previous two use cases in many ways, including (1) it is a real-world problem where an exact analytical expression is unknown; (2) the problem does not involve dynamics, rather, it is a regression problem on a static dataset; and (3) the dataset is not made of particles, but rather a grid of density that has been grouped and reduced to handmade features. Similarly, we do not know the dimensionality of interactions, should a linear latent space exist. We rely on our inductive bias to find the optimal dimensional of the problem, and then yield an interpretable model that performs better than existing analytical approximations. We refer our readers to our section 4.3 and the Appendix for further details.

## 4 Experiments & results

### 4.1 Newtonian dynamics

We train our Newtonian dynamics GNs on data for simple N-body systems with known force laws. We then apply our technique to recover the known force laws via the representations learned by the message function $\phi^e$.

**Data.** The dataset consists of N-body particle simulations in two and three dimensions, under different interaction laws. We used the following forces: (a) $1/r$ orbital force: $-m_1 m_2 \hat{r}/r$; (b) $1/r^2$

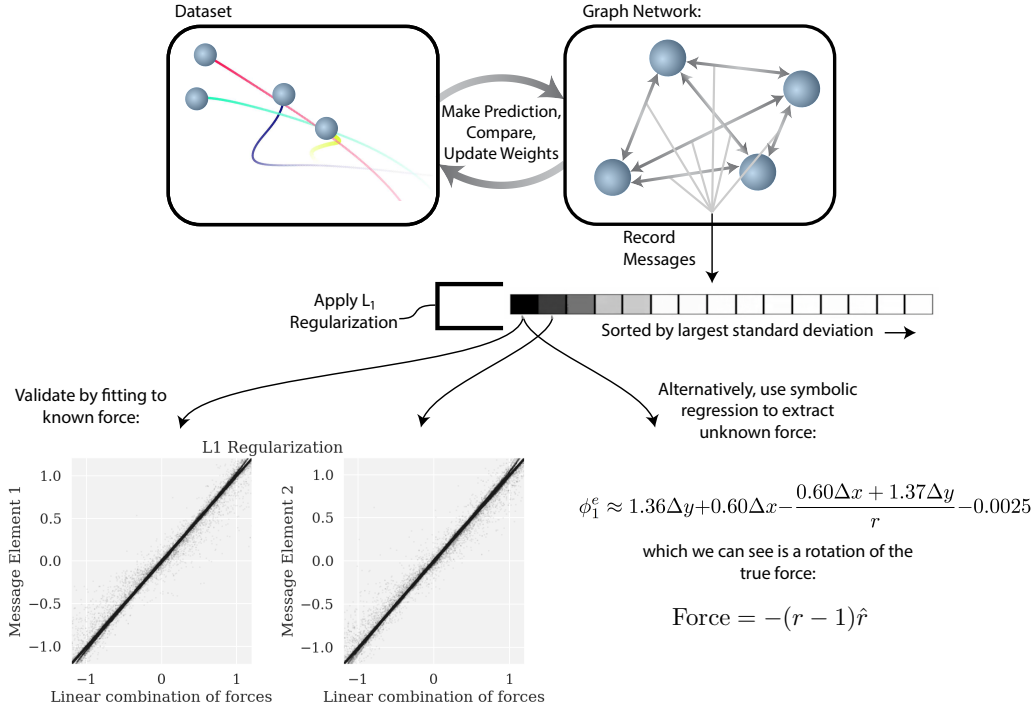

Figure 3: A diagram showing how we implement and exploit our inductive bias on GNs. A video of this figure during training can be seen by going to the URL `https://github.com/MilesCranmer/symbolic_deep_learning/blob/master/video_link.txt`.

orbital force $-m_1 m_2 \hat{r}/r^2$; (c) charged particles force $q_1 q_2 \hat{r}/r^2$; (d) damped springs with $|r-1|^2$ potential and damping proportional and opposite to speed; (e) discontinuous forces, $-\{0, r^2\}\hat{r}$, switching to 0 force for $r < 2$; and (f) springs between all particles, a $(r-1)^2$ potential. The simulations themselves contain masses and charges of 4 or 8 particles, with positions, velocities, and accelerations as a function of time. Further details of these systems are given in the appendix, with example trajectories shown in fig. 4.

**Model training.** The models are trained to predict instantaneous acceleration for every particle given the current state of the system. To investigate the importance of the size of the message representations for interpreting the messages as forces, we train our GN using 4 different strategies: 1. Standard, a GN with 100 message components; 2. Bottleneck, a GN with the number of message components matching the dimensionality of the problem (2 or 3); 3. $L_1$, same as "Standard" but using a $L_1$ regularization loss term on the messages with a weight of $10^{-2}$; and 4. KL same as "Standard" but regularizing the messages using the Kullback-Leibler (KL) divergence with respect to Gaussian prior. Both the $L_1$ and KL strategies encourage the network to find compact representations for the message vectors, using different regularizations. We optimize the mean absolute loss between the predicted acceleration and the true acceleration of each node. Additional training details are given in the appendix and found in the codebase.

**Performance comparison.** To evaluate the learned models, we generate a new dataset from a different random seed. We find that the model with $L_1$ regularization has the greatest prediction performance in most cases (see table 3). It is worth noting that the bottleneck model, even though it has the correct dimensionalty, performs worse than the model using $L_1$ regularization under limited training time. We speculate that this may connect to the lottery ticket hypothesis [50].

**Interpreting the message components.** As a first attempt to interpret the information in the message components, we pick the $D$ message features (where $D$ is the dimensionality of the simulation) with the highest variance (or KL divergence), and fit each to a linear combination of the

true force components. We find that while the GN trained in the Standard setting does not show strong correlations with force components (also seen in fig. 5), all other models for which the effective message size is constrained explicitly (bottleneck) or implicitly (KL or $L_1$) to be low dimensional yield messages that are highly correlated with the true forces (see table 1 which indicates the fit errors with respect to the true forces), with the model trained with $L_1$ regularization showing the highest correlations. An explicit demonstration that the messages in a graph network learn forces has not been observed before our work.

The messages in these models are thus explicitly interpretable as forces. The video at `https://github.com/MilesCranmer/symbolic_deep_learning/blob/master/video_link.txt` (fig. 3) shows a fit of the message components over time during training, showing how the model discovers a message representation that is highly correlated with a rotation of the true force vector in an unsupervised way.

| Sim. | Standard | Bottleneck | $L_1$ | KL |
|---|---|---|---|---|
| Charge-2 | 0.016 | 0.947 | 0.004 | 0.185 |
| Charge-3 | 0.013 | 0.980 | 0.002 | 0.425 |
| $r^{-1}$-2 | 0.000 | 1.000 | 1.000 | 0.796 |
| $r^{-1}$-3 | 0.000 | 1.000 | 1.000 | 0.332 |
| $r^{-2}$-2 | 0.004 | 0.993 | 0.990 | 0.770 |
| $r^{-2}$-3 | 0.002 | 0.994 | 0.977 | 0.214 |
| Spring-2 | 0.032 | 1.000 | 1.000 | 0.883 |
| Spring-3 | 0.036 | 0.995 | 1.000 | 0.214 |

Table 1: The $R^2$ value of a fit of a linear combination of true force components to the message components for a given model (see text). Numbers close to 1 indicate the messages and true force are strongly correlated. Successes/failures of force law symbolic regression are tabled in the appendix.

**Symbolic regression on the internal functions.** We now demonstrate symbolic regression to extract force laws from the messages, without using prior knowledge for each force's form. To do this, we record the most significant message component of $\phi^e$, which we refer to as $\phi_1^e$, over random samples of the training dataset. The inputs to the regression are $m_1, m_2, q_1, q_2, x_1, x_2, \ldots$ (mass, charge, x-position of receiving and sending node) as well as simplified variables to help the symbolic regression: e.g., $\Delta x$ for $x$ displacement, and $r$ for distance.

We then use *eureqa* to fit the $\phi_1^e$ to the inputs by minimizing the mean absolute error (MAE) over various analytic functions. Analogous to Occam's razor, we find the "best" algebraic model by asking *eureqa* to provide multiple candidate fits at different complexity levels (where complexity is scored as a function of the number and the type of operators, constants and input variables used), and select the fit that maximizes the fractional drop in mean absolute error (MAE) over the increase in complexity from the next best model: $(-\Delta \log(\text{MAE}_c)/\Delta c)$. From this, we recover many analytical expressions (this is tabled in the appendix) that are equivalent to the simulated force laws ($a, b$ indicate learned constants):

- Spring, 2D, $L_1$ (expect $\phi_1^e \approx (\mathbf{a} \cdot (\Delta x, \Delta y))(r - 1) + b$).
$$\phi_1^e \approx 1.36\Delta y + 0.60\Delta x - \frac{0.60\Delta x + 1.37\Delta y}{r} - 0.0025$$

- $1/r^2$, 3D, Bottleneck (expect $\phi_1^e \approx \frac{\mathbf{a} \cdot (\Delta x, \Delta y, \Delta z)}{r^3} + b$).
$$\phi_1^e \approx \frac{0.021\Delta x m_2 - 0.077\Delta y m_2}{r^3}$$

- Discontinuous, 2D, $L_1$ (expect $\phi_1^e \approx \text{IF}(r > 2, (\mathbf{a} \cdot (\Delta x, \Delta y, \Delta z))r, 0) + b$).
$$\phi_1^e \approx \text{IF}(r > 2, 0.15r\Delta y + 0.19r\Delta x, 0) - 0.038$$

Note that reconstruction does not always succeed, especially for training strategies other than $L_1$ or bottleneck models that cannot successfully find compact representations of the right dimensionality (see some examples in Appendix).

## 4.2 Hamiltonian dynamics

Using the same datasets from the Newtonian dynamics case study, we also train our "FlatHGN," with the Hamiltonian inductive bias, and demonstrate that we can extract scalar potential energies, rather than forces, for all of our problems. For example, in the case of charged particles, with expected potential ($\mathcal{H}_{\text{pair}} \approx \frac{aq_1 q_2}{r}$), symbolic regression applied to the learned message function yields[2]: $\mathcal{H}_{\text{pair}} \approx \frac{0.0019 q_1 q_2}{r}$.

It is also possible to fit the per-particle term $\mathcal{H}_{\text{self}}$, however, in this case the same kinetic energy expression is recovered for all systems. In terms of performance results, the Hamiltonian models are comparable to that of the $L_1$ regularized model across all datasets (See Supplementary results table).

Note that in this case, by design, the "FlatHGN" has a message function with a dimensionality of 1 to match the output of the Hamiltonian function which is a scalar, so no regularization is needed, as the message size is directly constrained to the right dimension.

## 4.3 Dark matter halos for cosmology

Now, one may ask: "will this strategy also work for general regression problems, non-trivial datasets, complex interactions, and unknown laws?" Here we give an example that satisfies all four of these concerns, using data from a gravitational simulation of the Universe.

Cosmology studies the evolution of the Universe from the Big Bang to the complex galaxies and stars we see today [51]. The interactions of various types of matter and energy drive this evolution. Dark Matter alone consists of $\approx 85\%$ of the total matter in the Universe [52, 53], and therefore is extremely important for the development of galaxies. Dark matter particles clump together and act as gravitational basins called "halos" which pull regular baryonic matter together to produce stars, and form larger structures such as filaments and galaxies. It is an important question in cosmology to predict properties of dark matter halos based on their "environment," which consist of other nearby dark matter halos. Here we study the following problem: how can we predict the excess amount of matter (in comparison to its surroundings, $\delta = \frac{\rho - \langle \rho \rangle}{\langle \rho \rangle}$) for a dark matter halo based on its properties and those of its neighboring dark matter halos?

A hand-designed estimator for the functional form of $\delta_i$ for halo $i$ might correlate $\delta$ with the mass of the same halo, $M_i$, as well as the mass within 20 distance units (we decide to use 20 as the smoothing radius): $\sum_{j \neq i}^{|\mathbf{r}_i - \mathbf{r}_j| < 20} M_j$. The intuition behind this scaling is described in [54]. Can we find a better equation that we can fit better to the data, using our methodology?

**Data, training and symbolic regression.** We study this problem with the open sourced N-body dark matter simulations from [49]. We choose the zeroth simulation in this dataset, at the final time step (current day Universe), which contains 215,854 dark matter halos. Each halo has mass $M_i$, position $\mathbf{r}_i$, and velocity $\mathbf{v}_i$. We also compute the smoothed overdensity $\delta_i$ at the location of the center of each halo. We convert this set of halos into a graph by connecting halos within fifty distance units (each distance unit is approximately 3 million light years long) of each other. This results in 30,437,218 directional edges between halos, or 71 neighbors per halo on average. We then attempt to predict $\delta_i$ for each halo with a GN. Training details are the same as for the Newtonian simulations, but we switch to 500 hidden units after hyperparameter tuning based on GN accuracy.

The GN trained with $L_1$ appears to have messages containing only 1 informative feature, so we extract message samples for this component of the messages over the training set for random pairs of halos, and node function samples for random receiving halos and their summed messages. The formula extracted by the algorithm is given in table 2 as "Best, with mass". The form of the formula is new and it captures a well-known relationship between halo mass and environment: bias-mass relationship. We refit the parameters in the formula on the original training data to avoid accumulated approximation error from the multiple levels of function fitting. We achieve a loss of 0.0882 where the hand-designed formula achieves a loss of 0.121. It is quite surprising that our formula extracted by our approach is able to achieve a better fit than the formula hand-designed by scientists.

| | Test | Formula | Summed Component | $\left\langle \left| \delta_i - \hat{\delta}_i \right| \right\rangle$ |
|---|---|---|---|---|
| Old | Constant | $\hat{\delta}_i = C_1$ | N/A | 0.421 |
| Old | Simple | $\hat{\delta}_i = C_1 + (C_2 + M_i C_3)e_i$ | $e_i = \sum_{j \neq i}^{\left| \mathbf{r}_i - \mathbf{r}_j \right| < 20} M_j$ | 0.121 |
| New | Best, without mass | $\hat{\delta}_i = C_1 + \frac{e_i}{C_2 + C_3 e_i \left| \mathbf{v}_i \right|}$ | $e_i = \sum_{j \neq i} \frac{C_4 + \left| \mathbf{v}_i - \mathbf{v}_j \right|}{C_5 + (C_6 \left| \mathbf{r_i} - \mathbf{r_j} \right|)^{C_7}}$ | 0.120 |
| New | Best, with mass | $\hat{\delta}_i = C_1 + \frac{e_i}{C_2 + C_3 M_i}$ | $e_i = \sum_{j \neq i} \frac{C_4 + M_j}{C_5 + (C_6 \left| \mathbf{r_i} - \mathbf{r_j} \right|)^{C_7}}$ | 0.0882 |

Table 2: A comparison of both known and discovered formulas for dark matter overdensity. $C_i$ indicates fitted parameters, which are given in the appendix.

The formula makes physical sense. Halos closer to the dark matter halo of interest should influence its properties more, and thus the summed function scales inversely with distance. Similar to the hand-designed formula, the overdensity should scale with the total matter density nearby, and we see this in that we are summing over mass of neighbors. The other differences are very interesting, and less clear; we plan to do detailed interpretation of these results in a future astrophysics study.

As a followup, we also calculated if we could predict the halo overdensity from only velocity and position information. This is useful because the most direct observational information available is in terms of halo velocities. We perform an identical analysis without mass information, and find a curiously similar formula. The relative speed between two neighbors can be seen as a proxy for mass, which is seen in table 2. This makes sense as a more massive object will have more gravity, accelerating falling particles near it to faster speeds. This formula is also new to cosmologists, and can in principle help push forward cosmological analysis.

**Symbolic generalization.** As we know that our physical world is well described by mathematics, we can use it as a powerful prior for creating new models of our world. Therefore, if we distill a neural network into a simple algebra, will the algebra generalize better to unseen data? Neural nets excel at learning in high-dimensional spaces, so perhaps, by combining both of these types of models, one can leverage the unique advantages of each. Such an idea is discussed in detail in [55].

Here we study this on the cosmology example by masking 20% of the data: halos which have $\delta_i > 1$. We then proceed through the same training procedure as before, learning a GN to predict $\delta$ with $L_1$ regularization, and then extracting messages for examples in the training set. Remarkably, we obtain a functionally identical expression when extracting the formula from the graph network on this subset of the data. We fit these constants to the same masked portion of data on which the graph network was trained. The graph network itself obtains an average error $\langle | \delta_i - \hat{\delta}_i | \rangle = 0.0634$ on the training set, and 0.142 on the out-of-distribution data. Meanwhile, the symbolic expression achieves 0.0811 on the training set, but 0.0892 on the out-of-distribution data. Therefore, for this problem, it seems a symbolic expression generalizes much better than the very graph neural network it was extracted from. This alludes back to Eugene Wigner's article: the language of simple, symbolic models is remarkably effective in describing the universe.

## 5   Conclusion

We have demonstrated a general approach for imposing physically motivated inductive biases on GNs and Hamiltonian GNs to learn interpretable representations, and potentially improved zero-shot generalization. Through experiment, we have shown that GN models which implement a bottleneck or $L_1$ regularization in the message passing layer, or a Hamiltonian GN flattened to pairwise and self-terms, can learn message representations equivalent to linear transformations of the true force vector or energy. We have also demonstrated a generic technique for finding an unknown force law from these models: symbolic regression is capable of fitting explicit equations to our trained model's message function. We repeated this for energies instead of forces via introduction of the "Flattened Hamiltonian Graph Network." Because GNs have more explicit substructure than their more homogeneous deep learning relatives (e.g., plain MLPs, convolutional networks), we can draw more fine-grained interpretations of their learned representations and computations. Finally, we have demonstrated our algorithm on a non-trivial dataset, and discovered a new law for cosmological dark matter.

*Acknowledgments:* Miles Cranmer would like to thank Christina Kreisch and Francisco Villaescusa-Navarro for assistance with the cosmology dataset; and Christina Kreisch, Oliver Philcox, and Edgar Minasyan for comments on a draft of the paper. The authors would like to thank the reviewers for insightful feedback that improved this paper. Shirley Ho and David Spergel's work is supported by the Simons Foundation.

Our code made use of the following Python packages: `numpy`, `scipy`, `sklearn`, `jupyter`, `matplotlib`, `pandas`, `torch`, `tensorflow`, `jax`, and `torch_geometric` [56, 57, 58, 59, 60, 61, 41, 62, 63, 42].

## 6 Broader impact

The approach we present has enormous potential for scientific research. Machine learning is traditionally focuses on prediction and pattern recognition, and science on understanding. The method we describe in this paper allows one to do both: understand, in an explicit algebraic language, how to model a dynamical system or how properties in a graph-like dataset relate to each other, using a deep neural network. Furthermore, our finding that a symbolic model extracted from components of a neural network will generalize better than the same neural network it was extracted from, is profound. We believe this interesting phenomenon is worth detailed followup studies to understand how this can be exploited more generally in machine learning.

In more specific terms:

- Scientists can now study graph-like datasets and potentially derive explicit symbolic expressions for interactions among the nodes. This could have a huge impact on learning new scientific models from data.

- Scientists can take deep learning models that can be easily trained and designed flexibly and use them to observe a system. Once the model is trained, scientists can easily take out a part of the model that they find important (such as the messages in the graph network) and use symbolic regression on that particular component to discover symbolic expressions governing properties of a physical system.

- More broadly, the technique we describe and then use graph networks to illustrate, can be applied to any trained networks, so long as the network has subcomponents that have an interpretable and modular form. It is not often obvious what components of a network would contain more important or interesting "laws" to extract, however, once that is identified, the approach presented in this paper can be used to extract an interpretable symbolic model.

## Footnotes

[1]`https://github.com/MilesCranmer/PySR`

[2]We have removed constant terms that don't depend on the position or momentum as those are just arbitrary offsets in the Hamiltonian which don't have an impact on the dynamics. See Appendix for more details.

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
