[Supplementary Material]

# Supplementary

## A  Model Implementation Details

Code for our implementation can be found at `https://github.com/MilesCranmer/symbolic_deep_learning`. Here we describe how one can implement our model from scratch in a deep learning framework. The main argument in this paper is that one can apply strong inductive biases to a deep learning model to simplify the extraction of a symbolic representation of the learned model. While we emphasize that this idea is general, in this section we focus on the specific Graph Neural Networks we have used as an example throughout the paper.

### A.1  Basic Graph Representation

We would like to use the graph $G = (V, E)$ to predict an updated graph $G' = (V', E)$. Our input dataset is a graph $G = (V, E)$ consisting of $N^v$ nodes with $L^v$ features each: $V = \{\mathbf{v}_i\}_{i=1:N^v}$, with each $\mathbf{v}_i \in \mathbb{R}^{L^v}$. The nodes are connected by $N^e$ edges: $E = \{(r_k, s_k)\}_{k=1:N^e}$, where $r_k, s_k \in \{1 : N^v\}$ are the indices for the receiving and sending nodes, respectively. We would like to use this graph to predict another graph $V' = \{\mathbf{v}'_i\}_{i=1:N^v}$, where each $\mathbf{v}'_i \in \mathbb{R}^{L^{v'}}$ is the node corresponding to $\mathbf{v}_i$. The number of features in these predicted nodes, $L^{v'}$, need not necessarily be the same as for the input nodes ($L^v$), though this could be the case for dynamical models where one is predicting updated states of particles. For more general regression problems, the number of output features is arbitrary.

**Edge model.**  The prediction is done in two parts. We create the first neural network, the edge model (or "message function"), to compute messages from one node to another: $\phi^e : \mathbb{R}^{L^v} \times \mathbb{R}^{L^v} \to \mathbb{R}^{L^{e'}}$. Here, $L^{e'}$ is the number of message features. In the bottleneck model, one sets $L^{e'}$ equal to the known dimension of the force, which is 2 or 3 for us. In our models, we set $L^{e'} = 100$ for the standard and $L_1$ models, and 200 for the KL model (which is described separately later on). We create $\phi^e$ as a multi-layer perceptron with ReLU activations and two hidden layers, each with 300 hidden nodes. The mapping is $\mathbf{e}'_k = \phi^e(\mathbf{v}_{r_k}, \mathbf{v}_{s_k})$ for all edges indexed by $k$ (i.e., we concatenate the receiving and sending node features).

**Aggregation.**  These messages are then pooled via element-wise summation for each receiving node $i$ into the summed message, $\bar{\mathbf{e}}'_i \in \mathbb{R}^{L^{e'}}$. This can be written as $\bar{\mathbf{e}}'_i = \sum_{k \in \{1:N^e | r_k=i\}} \mathbf{e}'_k$.

**Node model.**  We create a second neural network to predict the output nodes, $\mathbf{v}'_i$, for each $i$ from the corresponding summed message and input node. This net can be written as $\phi^v : \mathbb{R}^{L^v} \times \mathbb{R}^{L^{e'}} \to \mathbb{R}^{L^{v'}}$, and has the mapping: $\hat{\mathbf{v}}'_i = \phi^v(\mathbf{v}_i, \bar{\mathbf{e}}'_i)$, where $\hat{\mathbf{v}}'_i$ is the prediction for $\mathbf{v}'_i$. We also create $\phi^v$ as a

multi-layer perceptron with ReLU activations and two hidden layers, each with 300 hidden nodes. This model is then trained with the loss function as described later in this section.

**Summary.** We can write out our forward model for the bottleneck, standard, and $L_1$ models as:

$$\text{Input graph } G = (V, E) \text{ with}$$

nodes (e.g., positions of particles) $V = \{\mathbf{v}_i\}_{i=1:N^v}$; $\mathbf{v}_i \in \mathbb{R}^{L^v}$, and

edges (indices of connected nodes) $E = \{(r_k, s_k)\}_{k=1:N^e}$; $r_k, s_k \in \{1 : N^v\}$.

Compute messages for each edge: $\mathbf{e}'_k = \phi^e(\mathbf{v}_{r_k}, \mathbf{v}_{s_k})$,

$$\mathbf{e}'_k \in \mathbb{R}^{L^{e'}}, \text{ then}$$

$$\text{sum for each receiving node } i : \bar{\mathbf{e}}'_i = \sum_{k \in \{1:N^e | r_k = i\}} \mathbf{e}'_k,$$

$$\bar{\mathbf{e}}'_i \in \mathbb{R}^{L^{e'}}.$$

Compute output node prediction: $\hat{\mathbf{v}}'_i = \phi^v(\mathbf{v}_i, \bar{\mathbf{e}}'_i)$

$$\hat{\mathbf{v}}'_i \in \mathbb{R}^{L^{v'}}.$$

**Loss.** We jointly optimize the parameters in $\phi^v$ and $\phi^e$ via mini-batch gradient descent with Adam as the optimizer. Our total loss function for optimizing is:

$$\mathcal{L} = \mathcal{L}_v + \alpha_1 \mathcal{L}_e + \alpha_2 \mathcal{L}_n, \text{ where}$$

$$\text{the prediction loss is } \mathcal{L}_v = \frac{1}{N^v} \sum_{i \in \{1:N^v\}} |\mathbf{v}'_i - \hat{\mathbf{v}}'_i|,$$

$$\text{the message regularization is } \mathcal{L}_e = \frac{1}{N^e} \begin{cases} \sum_{k \in \{1:N^e\}} |\mathbf{e}'_k|, & \text{L}_1 \\ 0, & \text{Standard} \\ 0, & \text{Bottleneck} \end{cases},$$

$$\text{with the regularization constant } \alpha_1 = 10^{-2}, \text{ and the}$$

$$\text{regularization for the network weights is } \mathcal{L}_n = \sum_{l = \{1:N^l\}} |w_l|^2,$$

$$\text{with } \alpha_2 = 10^{-8},$$

where $\mathbf{v}'_i$ is the true value for the predicted node $i$. $w_l$ is the $l$-th network parameter out of $N^l$ total parameters. This implementation can be visualized during training in the video `https://github.com/MilesCranmer/symbolic_deep_learning`. During training, we also apply a random translation augmentation to all the particle positions to artificially generate more training data.

Next, we describe the KL variant of this model. Note that for the cosmology example in section 4.3, we use the $L_1$ model described above with 500 hidden nodes (found with coarse hyperparameter tuning to optimize accuracy) instead of 300, but other parameters are set the same.

## A.2 KL Model

The KL model is a variational version of the GN implementation above, which models the messages as distributions. We choose a normal distribution for each message component with a prior of $\mu = 0$, $\sigma = 1$. More specifically, the output of $\phi^e$ should now map to twice as many features as it is predicting a mean and variance, hence we set $L^{e'} = 200$. The first half of the outputs of $\phi^e$ now represent the means, and the second half of the outputs represent the log variance of a particular

message component. In other words,

$$\boldsymbol{\mu}'_k = \phi^e_{1:100}(\mathbf{v}_{r_k}, \mathbf{v}_{s_k}),$$
$$\boldsymbol{\sigma}'^2_k = \exp\big(\phi^e_{101:200}(\mathbf{v}_{r_k}, \mathbf{v}_{s_k})\big),$$
$$\mathbf{e}'_k \sim \mathcal{N}(\boldsymbol{\mu}'_k, \mathrm{diag}(\boldsymbol{\sigma}'^2_k)),$$
$$\bar{\mathbf{e}}'_i = \sum_{k \in \{1:N^e | r_k = i\}} \mathbf{e}'_k,$$
$$\hat{\mathbf{v}}'_i = \phi^v(\mathbf{v}_i, \bar{\mathbf{e}}'_i),$$

where $\mathcal{N}$ is a multinomial Gaussian distribution. Every time the graph network is run, we calculate the mean and log variance of messages, sample each message once to calculate $\mathbf{e}'_k$, and pass those samples through a sum to compute a sample of $\bar{\mathbf{e}}'_i$ and then pass that value through the edge function to compute a sample of $\hat{\mathbf{v}}'_i$. The loss is calculated normally, except for $\mathcal{L}_e$, which becomes the KL divergence with respect to our Gaussian prior of $\mu = 0$, $\sigma = 1$:

$$\mathcal{L}_e = \frac{1}{N^e} \sum_{k = \{1:N^e\}} \sum_{j = \{1:L^{e'}/2\}} \frac{1}{2}\left(\mu'^2_{k,j} + \sigma'^2_{k,j} - \log\big(\sigma'^2_{k,j}\big)\right),$$

with $\alpha_1 = 1$ (equivalent to $\beta = 1$ for the loss of a $\beta$-Variational Autoencoder; simply the standard VAE). The KL-divergence loss also encourages sparsity in the messages $\mathbf{e}'_k$ similar to the $\mathrm{L}_1$ loss. The difference is that here, an uninformative message component will have $\mu = 0, \sigma = 1$ (a KL of 0) rather than a small absolute value. We train the networks with a decaying learning schedule as given in the example code.

### A.3  Constraining Information in the Messages

The hypothesis which motivated our graph network inductive bias is that if one minimizes the dimension of the vector space used by messages in a GN, the components of message vectors will learn to be linear combinations of the true forces (or equivalent underlying summed function) for the system being learned. The key observation is that $\mathbf{e}'_k$ could learn to correspond to the true force vector imposed on the $r_k$-th body due to its interaction with the $s_k$-th body.

Here, we sketch a rough mathematical explanation of our hypothesis that we will reconstruct the true force in the graph network given our inductive biases. Newtonian mechanics prescribes that force vectors, $\mathbf{f}_k \in \mathcal{F}$, can be summed to produce a net force, $\sum_k \mathbf{f}_k = \bar{\mathbf{f}} \in \mathcal{F}$, which can then be used to update the dynamics of a body. Our model uses the $i$-th body's pooled messages, $\bar{\mathbf{e}}'_i$ to update the body's state via $\mathbf{v}'_i = \phi^v(\mathbf{v}_i, \bar{\mathbf{e}}'_i)$. If we assume our GN is trained to predict accelerations perfectly for any number of bodies, this means (ignoring mass) that $\bar{\mathbf{f}}_i = \sum_{r_k = i} \mathbf{f}_k = \phi^v(\mathbf{v}_i, \sum_{r_k = i} \mathbf{e}'_k) = \phi^v(\mathbf{v}_i, \bar{\mathbf{e}}'_i)$. Since this is true for any number of bodies, we also have the result for a single interaction: $\bar{\mathbf{f}}_i = \mathbf{f}_{k, r_k = i} = \phi^v(\mathbf{v}_i, \mathbf{e}'_{k, r_k = i}) = \phi^v(\mathbf{v}_i, \bar{\mathbf{e}}'_i)$. Thus, we can substitute this expression into the multi-interaction case: $\sum_{r_k = i} \phi^v(\mathbf{v}_i, \mathbf{e}'_k) = \phi^v(\mathbf{v}_i, \bar{\mathbf{e}}'_i) = \phi^v(\mathbf{v}_i, \sum_{r_k = i} \mathbf{e}'_k)$. From this relation, we see that $\phi^v$ has to be a linear transformation conditioned on $\mathbf{v}_i$. Therefore, for cases where $\phi^v(\mathbf{v}_i, \bar{\mathbf{e}}'_i)$ is invertible in $\bar{\mathbf{e}}'_i$ (which becomes true when $\bar{\mathbf{e}}'_i$ is the same dimension as the output of $\phi^v$), we can write $\mathbf{e}'_k = (\phi^v(\mathbf{v}_i, \cdot))^{-1}(\mathbf{f}_k)$, which is also a linear transform, meaning that the message vectors are linear transformations of the true forces when $L^{e'}$ is equal to the dimension of the forces.

If the dimension of the force vectors (or what the minimum dimension of the message vectors "should" be) is unknown, one can encourage the messages to be sparse by applying $\mathrm{L}_1$ or Kullback-Leibler regularizations to the messages in the GN. The aim is for the messages to learn the minimal vector space required for the computation automatically. This is a more mathematical explanation of why the message features are linear combinations of the force vectors, when our inductive bias of a bottleneck or sparse regularization is applied. We emphasize that this is a new contribution: never before has previous work explicitly identified the forces in a graph network.

**General Graph Neural Networks.**  In all of our models here, we assume the dataset does not have edge-specific features, such as a different coupling constants between different particles, but these could be added by concatenating edge features to the receiving and sending node input to $\phi^e$. We also assume there are no global properties. The graph neural network is described in general form in [4]. All of our techniques are applicable to the general form: one would approximate $\phi^e$ with a symbolic model with included input edge parameters, and also fit the global model, denoted $\phi^u$.

## A.4 Flattened Hamiltonian Graph Network.

As part of this study, we also consider an alternate dynamical model that is described by a linear latent space other than force vectors. In the Hamiltonian formalism of classical mechanics, energies of pairwise interactions and kinetic and potential energies of particles are pooled into a global energy value, $\mathcal{H}$, which is a scalar. We label pairwise interaction energy $\mathcal{H}_{\text{pair}}$ and the energy of individual particles as $\mathcal{H}_{\text{self}}$. Thus, using our previous graph notation, we can write the total energy of a system as:

$$\mathcal{H} = \sum_{i=1:N^v} \mathcal{H}_{\text{self}}(\mathbf{v}_i) + \sum_{k \in \{1:N^e\}} \mathcal{H}_{\text{pair}}(\mathbf{v}_{r_k}, \mathbf{v}_{s_k}). \tag{1}$$

For particles interacting via gravity, this would be

$$\mathcal{H} = \sum_i \frac{p_i^2}{2m_i} - \frac{1}{2} \sum_{i \neq j} \frac{m_i m_j}{|\mathbf{r}_i - \mathbf{r}_j|}, \tag{2}$$

where $\mathbf{p}_i, m_i, \mathbf{r}_i$ indicates the momentum, mass, and position of particle $i$, respectively, and we have set the gravitational constant to 1. Following [45, 44], we could model $\mathcal{H}$ as a neural network, and apply Hamilton's equations to create a dynamical model. More specifically, as in [44], we can predict $\mathcal{H}$ as the global property of a GN (this is called a Hamiltonian Graph Network or HGN). However, energy, like forces in Cartesian coordinates, is a summed quantity. In other words, energy is another "linear latent space" that describes the dynamics.

Therefore, we argue that an HGN will be more interpretable if we explicitly sum up energies over the system, rather than compute $\mathcal{H}$ as a global property of a GN. Here, we introduce the "Flattened Hamiltonian Graph Network," or "FlatHGN", which uses eq. (1) to construct a model that works on a graph. We set up two Multi-Layer Perceptrons (MLPs), one for each node:

$$\mathcal{H}_{\text{self}} : \mathbb{R}^{L^v} \to \mathbb{R}, \tag{3}$$

and one for each edge:

$$\mathcal{H}_{\text{pair}} : \mathbb{R}^{L^v} \times \mathbb{R}^{L^v} \to \mathbb{R}. \tag{4}$$

Note that the derivatives of $\mathcal{H}$ now propagate through the pool, e.g.,

$$\frac{\partial \mathcal{H}(V)}{\partial \mathbf{v}_i} = \frac{\partial \mathcal{H}_{\text{self}}(\mathbf{v}_i)}{\partial \mathbf{v}_i} + \sum_{r_k = i} \frac{\partial \mathcal{H}_{\text{pair}}(\mathbf{e}_k, \mathbf{v}_{r_k}, \mathbf{v}_{s_k})}{\partial \mathbf{v}_i}$$
$$+ \sum_{s_k = i} \frac{\partial \mathcal{H}_{\text{pair}}(\mathbf{e}_k, \mathbf{v}_{r_k}, \mathbf{v}_{s_k})}{\partial \mathbf{v}_i}. \tag{5}$$

This model is similar to the Lagrangian Graph Network proposed in [47]. Now, should this FlatHGN learn energy functions such that we can successfully model the dynamics of the system with Hamilton's equations, we would expect that $\mathcal{H}_{\text{self}}$ and $\mathcal{H}_{\text{pair}}$ should be analytically similar to parts of the true Hamiltonian. Since we have broken the traditional HGN into a FlatHGN, we now have pairwise and self energies, rather than a single global energy, and these are simpler to extract and interpret. This is a similar inductive bias to the GN we introduced previously. To train a FlatHGN, one can follow our strategy above, with the output predictions made using Hamilton's equations applied to our $\mathcal{H}$. One difference is that we also regularize $\mathcal{H}_{\text{pair}}$, since it is degenerate with $\mathcal{H}_{\text{self}}$ in that it can pick up self energy terms.

## B   Simulations

Our simulations for sections 4.1 and 4.2 were written using the JAX library (`https://github.com/google/jax`) so that we could easily vectorize computations over the entire dataset of 10,000 simulations. Example "long exposures" for each simulation in 2D are shown in fig. 4. To create each simulation, we set up the following potentials between two particles, 1 (receiving) and 2 (sending). Here, $r'_{12}$ is the distance between two particles plus 0.01 to prevent singularities. For particle $i$, $m_i$ is the mass, $q_i$ is the charge, $n$ is the number of particles in the simulation, $\mathbf{r}_i$ is the position of a

Figure 4: Examples of a selection of simulations, for 4 nodes and two dimensions. Decreasing transparency shows increasing time, and size of points shows mass.

particle, and $\dot{\mathbf{r}}_i$ is the velocity of a particle.

$$1/r^2 : \; U_{12} = -m_1 m_2 / r'_{12}$$
$$1/r : \; U_{12} = m_1 m_2 \log(r'_{12})$$
$$\text{Spring} : \; U_{12} = (r'_{12} - 1)^2$$
$$\text{Damped} : \; U_{12} = (r'_{12} - 1)^2 + \mathbf{r}_1 \cdot \dot{\mathbf{r}}_1 / n$$
$$\text{Charge} : \; U_{12} = q_1 q_2 / r'_{12}$$
$$\text{Dicontinuous} : \; U_{12} = \left\{ \begin{array}{ll} 0, & r'_{12} < 2 \\ (r'_{12} - 1)^2, & r'_{12} \geq 2 \end{array} \right.$$

All variables lack units. Here, $m_i$ is sampled from a log-normal distribution with $\mu = 0, \sigma = 1$. Each component of $\mathbf{r}_i$ and $\dot{\mathbf{r}}_i$ is randomly sampled from a normal distribution with $\mu = 0, \sigma = 1$. $q_i$ is randomly drawn from a set of two elements: $\{-1, 1\}$, representing charge. The acceleration of a given particle is then

$$\ddot{\mathbf{r}}_i = -\frac{1}{m_i} \sum_j \nabla_{\mathbf{r}_i} U_{ij}. \tag{6}$$

This is integrated over 1000 time steps of a fixed step size for a given random initial configuration using an adaptive RK4 integrator. The step size varies for each simulation due to the differences in scale. It is: 0.005 for $1/r$, 0.001 for $1/r^2$, 0.01 for Spring, 0.02 for Damped, 0.001 for Charge, and 0.01 for Discontinuous. Each simulation is performed in two and three dimensions, for 4 and 8 bodies. We store these simulations on disk. For training, the simulations for the particular problem being studied are loaded, and each instantaneous snapshot of each simulation is converted to a fully connected graph, with the predicted property (nodes of $V'$, see appendix A) being the acceleration of the particles at that snapshot.

The test loss of each model trained on each simulation set is given in table 3.

As described in the text (and visualized in the drive video), we can fit linear combinations of the true force components to each of the significant features of a message vector. This fit is summarized by table 1, and the fit itself is visualized in fig. 5 for various models on the 2D spring simulation.

| Sim. | Standard | Bottleneck | $L_1$ | KL | FlatHGN |
|---|---|---|---|---|---|
| Charge-2 | **49** | 50 | 52 | 60 | 55 |
| Charge-3 | 1.2 | 0.99 | **0.94** | 4.2 | 3.5 |
| Damped-2 | **0.30** | 0.33 | **0.30** | 1.5 | 0.35 |
| Damped-3 | 0.41 | 0.45 | **0.40** | 3.3 | 0.47 |
| Disc.-2 | 0.064 | 0.074 | **0.044** | 1.8 | 0.075 |
| Disc.-3 | 0.20 | 0.18 | **0.13** | 4.2 | 0.14 |
| $r^{-1}$-2 | 0.077 | 0.069 | 0.079 | 3.5 | **0.05** |
| $r^{-1}$-3 | 0.051 | 0.050 | 0.055 | 3.5 | **0.017** |
| $r^{-2}$-2 | 1.6 | 1.6 | **1.2** | 9.3 | 1.3 |
| $r^{-2}$-3 | 4.0 | 3.6 | 3.4 | 9.8 | **2.5** |
| Spring-2 | 0.047 | 0.046 | 0.045 | 1.7 | **0.016** |
| Spring-3 | 0.11 | 0.11 | 0.090 | 3.8 | **0.010** |

Table 3: Test prediction losses for each model on each dataset in two and three dimensions. The training was done with the same batch size, schedule, and number of epochs.

Figure 5: The most significant message components of each model compared with a linear combination of the force components: this example, the spring simulation in 2D with eight nodes for training. These plots demonstrate that the GN's messages have learned to be linear transformations of the vector components of the true force, in this case a springlike force, after applying an inductive bias to the messages.

## C   Symbolic Regression Details

After training a model on each simulation, we convert a deep learning model to a symbolic expression by approximating subcomponents of the model with symbolic regression, over observed inputs and outputs. For our aforementioned GNN implementation, we can record the outputs of $\phi^e$ and $\phi^v$ for various data points in the training set.

For models other than the bottleneck and Hamiltonian model (where we explicitly limit the features) we calculate the most significant output features of $\phi^e$ (we also refer to the output features as "message components"). For the $L_1$ and standard model, this is done by sorting the message components with the largest standard deviation; the most significant feature is the one with the largest standard deviation, which are the features we study. For the KL model, we consider the feature with the largest KL-divergence: $\mu^2 + \sigma^2 - \log(\sigma^2)$. These features are the ones we consider to be containing information used by the GN, so are the ones we fit symbolic expressions to.

As an example, here we fit the most significant feature, which we refer to as $\phi_1^e$, over random examples of the training dataset. We do this for the particle simulations in section 4.1. The inputs to the actual $\phi_1^e$ neural network are: $m_1, m_2, q_1, q_2, x_1, x_2, \dots$ (mass, charge, and Cartesian positions of receiving

and sending node), leaving us with many examples of $(m_1, m_2, q_1, q_2, x_1, x_2, \ldots, \phi_1^e)$. We would like to fit a symbolic expression to map $(m_1, m_2, q_1, q_2, x_1, x_2, \ldots) \rightarrow \phi_1^e$. To simplify things for this symbolic model, we convert the input position variables to a more interpretable format: $\Delta x = x_2 - x_1$ for $x$ displacement, likewise for $y$ (and $z$, if it is a 3D simulation), and $r = \sqrt{\Delta x^2 + \Delta y^2 (+\Delta z^2)}$ for distance.

We then pass these $(m_1, m_2, q_1, q_2, \Delta x, \Delta y, (\Delta z, )r, \phi_1^e)$ examples (we take 5000 examples for each of our tests) to *eureqa*, and ask it to fit $\phi_1^e$ as a function of the others by minimizing the mean absolute error (MAE). We allow it to use the operators $+, -, \times, /, >, <, \wedge, \exp, \log, \mathrm{IF}(\cdot, \cdot, \cdot)$ as well as real constants in its solutions. We score complexity by counting the number of occurrences of each operator, constant, and input variable. We weight $\wedge, \exp, \log, \mathrm{IF}(\cdot, \cdot, \cdot)$ as three times the other operators, since these are more complex operations. *eureqa* outputs the best equation at each complexity level, denoted by $c$. Example outputs are shown in table 4 for the $1/r$ and $1/r^2$ simulations. We select a formula from this list by taking the one that maximizes the fractional drop in mean absolute error (MAE) over an increase in complexity from the next best model. This is analogous to Occam's Razor: we jointly optimize for simplicity and accuracy of the model. The objective itself can be written as maximizing $(-\Delta \log(\mathrm{MAE}_c)/\Delta c)$ over the best model at each maximum complexity level, and is schematically illustrated in fig. 6. We find experimentally that this score produces the best-recovered solutions in a variety of tests on different generating equations.

Following the process of fitting analytic equations to the messages, we fit a single analytic expression to model $\phi_1^e$ as a function of the simplified input variables. We recover many analytical expressions that were used to generate the data, examples of which are listed below ($a, b$ indicate learned constants):

- Spring, 2D, L$_1$ (expect $\phi_1^e \approx (\mathbf{a} \cdot (\Delta x, \Delta y))(r - 1) + b$).

$$\phi_1^e \approx 1.36\Delta y + 0.60\Delta x - \frac{0.60\Delta x + 1.37\Delta y}{r} - 0.0025$$

- $1/r^2$, 3D, Bottleneck (expect $\phi_1^e \approx \frac{\mathbf{a} \cdot (\Delta x, \Delta y, \Delta z)}{r^3} + b$).

$$\phi_1^e \approx \frac{0.021\Delta x m_2 - 0.077\Delta y m_2}{r^3}$$

- Discontinuous, 2D, L$_1$ (expect $\phi_1^e \approx \mathrm{IF}(r > 2, (\mathbf{a} \cdot (\Delta x, \Delta y, \Delta z))r, 0) + b$).

$$\phi_1^e \approx \mathrm{IF}(r > 2, 0.15r\Delta y + 0.19r\Delta x, 0) - 0.038$$

**Examples of failed reconstructions.** Note that reconstruction does not always succeed, especially for training strategies other than L$_1$ or bottleneck models that cannot successfully find compact representations of the right dimensionality. We demonstrate some failed examples below:

- Spring, 3D, KL (expect $\phi_1^e \approx (\mathbf{a} \cdot (\Delta x, \Delta y, \Delta z))(r - 1) + b$).

$$\phi_1^e \approx 0.57\Delta y + 0.32\Delta z$$

- $1/r$, 3D, Standard (expect $\phi_1^e \approx \frac{\mathbf{a} \cdot (\Delta x, \Delta y, \Delta z)}{r^2} + b$).

$$\phi_1^e \approx \frac{0.041 + m_2 \mathrm{IF}(\Delta z > 0, 0.021, 0.067)}{r}$$

We do not attempt to make any general statements about when symbolic regression applied to the message components will fail or succeed in extracting the true law. Simply, we show that it is possible, for a variety of physical systems, and argue that reconstruction is more likely by the inclusion of a strong inductive bias in the network.

A full table of successes and failures in reconstructing the force law over the different n-body experiments is given in table 5. While the equations given throughout the paper were generated with *eureqa*, to create this table in particular, we switched from *eureqa* to *PySR*. This is because *PySR* allows us to configure a controlled experiment with fixed hyperparameters and total mutation steps for each force law, whereas Eureqa makes these controls inaccessible. However, given enough training time, we found *eureqa* and *PySR* produced equivalent results for equations at this simplicity level.

| Solutions extracted for the 2D $1/r^2$ Simulation | MAE | Complexity |
|---|---|---|
| $\phi_1^e = 0.162 + (5.62 + 20.3m_2\Delta x - 153m_2\Delta y)/r^3$ | 17.954713 | 22 |
| $\phi_1^e = (6.07 + 19.9m_2\Delta x - 154m_2\Delta y)/r^3$ | 18.400224 | 20 |
| $\phi_1^e = (3.61 + 20.9\Delta x - 154m_2\Delta y)/r^3$ | 42.323236 | 18 |
| $\phi_1^e = (31.6\Delta x - 152m_2\Delta y)/r^3$ | 69.447467 | 16 |
| $\phi_1^e = (2.78 - 152m_2\Delta y)/r^3$ | 131.42547 | 14 |
| $\phi_1^e = -142m_2\Delta y/r^3$ | 160.31243 | 12 |
| $\phi_1^e = -184\Delta y/r^2$ | 913.83751 | 8 |
| $\phi_1^e = -7.32\Delta y/r$ | 1520.9493 | 6 |
| $\phi_1^e = -0.282m_2\Delta y$ | 1551.3437 | 5 |
| $\phi_1^e = -0.474\Delta y$ | 1558.9756 | 3 |
| $\phi_1^e = 0.0148$ | 1570.0905 | 1 |

| Solutions extracted for the 2D $1/r$ Simulation | MAE | Complexity |
|---|---|---|
| $\phi_1^e = (4.53m_2\Delta y - 1.53\Delta x - 15.0m_2\Delta x)/r^2 - 0.209$ | 0.37839388 | 22 |
| $\phi_1^e = (4.58m_2\Delta y - \Delta x - 15.2m_2\Delta x)/r^2 - 0.227$ | 0.38 | 20 |
| $\phi_1^e = (4.55m_2\Delta y - 15.5m_2\Delta x)/r^2 - 0.238$ | 0.42 | 18 |
| $\phi_1^e = (4.59m_2\Delta y - 15.5m_2\Delta x)/r^2$ | 0.46575519 | 16 |
| $\phi_1^e = (10.7\Delta y - 15.5m_2\Delta x)/r^2$ | 2.48 | 14 |
| $\phi_1^e = (\Delta y - 15.6m_2\Delta x)/r^2$ | 6.96 | 12 |
| $\phi_1^e = -15.6m_2\Delta x/r^2$ | 7.93 | 10 |
| $\phi_1^e = -34.8\Delta x/r^2$ | 31.17 | 8 |
| $\phi_1^e = -8.71\Delta x/r$ | 68.345174 | 6 |
| $\phi_1^e = -0.360m_2\Delta x$ | 85.743106 | 5 |
| $\phi_1^e = -0.632\Delta x$ | 93.052677 | 3 |
| $\phi_1^e = -\Delta x$ | 96.708906 | 2 |
| $\phi_1^e = -0.303$ | 103.29053 | 1 |

Table 4: Results of using symbolic regression to fit equations to the most significant (see text) feature of $\phi^e$, denoted $\phi_1^e$, for the $1/r^2$ (top) and $1/r$ (bottom) force laws, extracted from the bottleneck model. We expect to see $\phi_1^e \approx \frac{\mathbf{a} \cdot (\Delta x, \Delta y, \Delta z)}{r^\alpha} + b$, for arbitrary $\mathbf{a}$ and $b$, and $\alpha = 2$ for the $1/r$ simulation and $\alpha = 3$ for the $1/r^2$ simulation, which is approximately what we recover. The row with a gray background has the largest fractional drop in mean absolute error in their tables, which according to our parametrization of Occam's razor, represents the best model. This demonstrates a technique for learning an unknown "force law" with a constrained graph neural network.

**Pure *eureqa* experiment**  To demonstrate that *eureqa* by itself is not capable of finding many of the equations considered from the raw high-dimensional dataset, we ran it on the simulation data without our GN's factorization of the problem, giving it the features of every particle. As expected, even after convergence, it cannot find meaningful equations; all of the solutions it provides for the n-body system are very poor fits. One such example of an equation, for the acceleration of particle 2 along the $x$ direction in a 6-body system under a $1/r^2$ force law, is:

$$\ddot{x}_2 = \frac{0.426}{367y_4 - 1470} + \frac{2.88 \times 10^5 x_1}{2.08 \times 10^3 y_4 + 446y_4^2} - 5.98 \times 10^{-5}x_6 - 109x_1,$$

where the indices refer to particle number. Despite *eureqa* converging, this equation is evidently meaningless and achieves a poor fit to the data. Thus, we argue that raw symbolic regression is intractable for the problems we consider, and only after factorization with a neural network do these problems become feasible for symbolic regression.

**Discovering potentials using FlatHGN.**  Lastly, we also show an example of a successful reconstruction of a pairwise Hamiltonian from data. We treat the $\mathcal{H}_{\text{pair}}$ just as we would $\phi_1^e$, and fit it to data. The one difference here is that there are potential $\mathcal{H}_{\text{pair}}$ values offset by a constant function of the non-dynamical parameters (fixed properties like mass) which still produce the correct dynamics, since only the derivatives of $\mathcal{H}_{\text{pair}}$ are used. Thus, we cannot simply fit a linear transformation of the true $\mathcal{H}_{\text{pair}}$ to data to verify it has learned our generating equation: we must rely on symbolic regression

Figure 6: A plot of the data for the $1/r$ simulation in table 4, indicating mean absolute error versus complexity in the top plot and fractional drop in mean absolute error over the next-best model in the bottom plot. As indicated, we take the largest drop in log-loss over a single increase in complexity as the chosen model—it is our parametrization of Occam's Razor.

| Sim. | Standard | Bottleneck | $L_1$ | KL |
|---|:---:|:---:|:---:|:---:|
| Charge-2 | ✗ | ✓ | ✗ | ✗ |
| Charge-3 | ✗ | ✓ | ✗ | ✗ |
| $r^{-1}$-2 | ✗ | ✓ | ✓ | ✓ |
| $r^{-1}$-3 | ✗ | ✓ | ✓ | ✓ |
| $r^{-2}$-2 | ✗ | ✓ | ✓ | ✗ |
| $r^{-2}$-3 | ✗ | ✓ | ✓ | ✗ |
| Spring-2 | ✗ | ✓ | ✓ | ✓ |
| Spring-3 | ✗ | ✓ | ✓ | ✓ |

Table 5: Success/failure of a reconstruction of the force law by symbolic regression, corresponding to the values in table 1.

to extract the full functional form. We follow the same procedure as before, and successfully extract the potential for a charge simulation:

$$\mathcal{H}_{\text{pair}} \approx \frac{0.0019 q_1 q_2}{r} - 0.0112 - 0.00143 q_1 - 0.00112 q_1 q_2,$$

where we expect $\mathcal{H}_{\text{pair}} \approx a \frac{q_1 q_2}{r} + f(q_1, q_2, m_1, m_2)$, for constant $a$ and arbitrary function $f$, which shows that the neural network has learned the correct form of the Hamiltonian.

**Hyperparameters.** Since the hyperparameters used internally by *eureqa* are opaque and not tunable, here we discuss the parameters used in *PySR* [25], which are common among many symbolic regression tools. At a given step of the training, there is a set of active equations in the "population". The number of active equations is a tunable hyperparameter, and is related to the diversity of the discovered equations, as well as the number of compute cores being used. The max size of equations controls the maximum complexity considered, and can be controlled to prevent the algorithm from wasting cycles on over-complicated equations. The operators used in the equations depends on the specific problem considered, and is another hyperparameter specified by the user. Next, there is a set of tunable probabilities associated with each mutation: how frequently to mutate an operator into a different operator, add an operator with arguments, replace an operator and its arguments with a constant, and so on. In some approaches such as with *PySR*, the best equations found over the course of training are randomly reintroduced back into the population. The frequency at which this occurs is controlled by another hyperparameter.

## D   Video Demonstration and Code

We include a video demonstration of the central ideas of our paper at `https://github.com/MilesCranmer/symbolic_deep_learning`. It shows the message components of a graph network converging to be equal to a linear combination of the force components when $L_1$ regularization is applied. Time in each clip of the video is correlated with training epoch. In this video, the top left corner of the fully revealed plot corresponds to a single test simulation that is 300 time steps long. Four particles of different masses are initiated with random positions and velocities, and evolved according to the potential of a spring with an equilibrium position of 1: $(r-1)^2$, where $r$ is the distance between two particles. The evaluation trajectories are shown on the right, with the gray particles indicating the true locations. The 15 largest message components in terms of standard deviation over a test set are represented in a sorted list below the graph network in gray, where darker color corresponds to a larger standard deviation. Since we apply $L_1$ regularization to the messages, we expect this list to grow sparser over time, which it does. Of these messages, the two largest components are extracted, and each is fit to a separate linear combination of the true force components (bottom left). A better fit to the true force components — indicating that the messages represent the force — are indicated by dots (each dot is a single message) that lie closer along the $y = x$ line in the bottom middle two scatter plots.

As can be seen in the video, as the messages grow increasingly sparse, the messages eventually converge to be almost exactly linear combinations of the true forces. Finally, once the loss is converged, we also fit symbolic regression to the largest message component. The video was created using the same training procedure as used in the rest of the paper. The dataset that the $L_1$ model was trained on is the 4-node Spring-2. Finally, we include the full code required to generate the animated clips in the above figure. This code contains all of the models and simulators used in the paper, along with the default training parameters. This code can also be accessed in the drive.

## E   Cosmological Experiments

For the cosmological data graph network, we do a coarse hyperparameter tuning based on predictions of $\delta_i$ and select a GN with 500 hidden units, two hidden layers per node function and message function. We choose 100 message dimensions as before. We keep other hyperparameters the same as before: $L_1$ regularization with a regularization scale of $10^{-2}$.

Remarkably, the vector space discovered by this graph network is 1 dimensional. This is indicated by the fact that only one message component has standard deviation of about $10^{-2}$ and all other 99 components have a standard deviation of under $10^{-8}$. This suggests that the $\delta_i$ prediction is a sum over some function of the center halo and each neighboring halo. Thus, we can rewrite our model as a sum over a function $\phi_1^e$ which takes the central halo and each neighboring halo, and passes it to $\phi^v$ which predicts $\delta_i$ given the central halo properties.

**Best-fit parameters.** We list best-fit parameters for the discovered models in the paper in table 6. The functional forms were extracted from the GN by approximating both $\phi_1^e$ and $\phi^v$ over training data with a symbolic regression and then analytically composing the expressions. Although the symbolic regression fits constants itself, this accumulates error from the two levels of approximation (graph net

| | Test | Formula | Summed Component | $\left\langle \left\lvert \delta_i - \hat{\delta}_i \right\rvert \right\rangle$ |
|---|---|---|---|---|
| Old | Constant | $\hat{\delta}_i = C_1$ | N/A | 0.421 |
| Old | Simple | $\hat{\delta}_i = C_1 + (C_2 + M_i C_3)e_i$ | $e_i = \sum_{j\neq i}^{\lvert \mathbf{r}_i - \mathbf{r}_j \rvert < 20} M_j$ | 0.121 |
| New | Best, without mass | $\hat{\delta}_i = C_1 + \frac{e_i}{C_2 + C_3 e_i \lvert \mathbf{v}_i \rvert}$ | $e_i = \sum_{j\neq i} \frac{C_4 + \lvert \mathbf{v}_i - \mathbf{v}_j \rvert}{C_5 + (C_6 \lvert \mathbf{r_i} - \mathbf{r_j} \rvert)^{C_7}}$ | 0.120 |
| New | Best, with mass | $\hat{\delta}_i = C_1 + \frac{e_i}{C_2 + C_3 M_i}$ | $e_i = \sum_{j\neq i} \frac{C_4 + M_j}{C_5 + (C_6 \lvert \mathbf{r_i} - \mathbf{r_j} \rvert)^{C_7}}$ | 0.0882 |

| Test | Best-fit Parameters |
|---|---|
| Simple | $C_1 = 0.415$ |
| Traditional | $C_1 = -0.0376, C_2 = 0.0529, C_3 = 0.000927$ |
| Best, without mass | $C_1 = -0.199, C_2 = 1.31, C_3 = 0.027,$ $C_4 = 1.54, C_5 = 50.165, C_6 = 18.94, C_7 = 13.21$ |
| Best, with mass | $C_1 = -0.156, C_2 = 3.80, C_3 = 0.0809,$ $C_4 = 0.438, C_5 = 7.06, C_6 = 15.5, C_7 = 20.3$ |
| Best, with mass and cutoff* | $C_1 = -0.149, C_2 = 3.77, C_3 = 0.0789,$ $C_4 = 0.442, C_5 = 7.09, C_6 = 15.5, C_7 = 21.3$ |

Table 6: Best-fit parameters for the functional forms used to estimate the overdensity of dark matter halos. The functional forms are given in the upper table for reference. *Here we use the same formula as "Best, with mass," since we found an equivalent formula by only looking at the 80% chunk of the data. The constants in that functional form are also fit by only training on that fraction of the data.

to data, symbolic regression to graph net). Thus, we take out the functional forms as given in table 6, and refit the parameters directly to the training data. This results in the parameters given, which are used to calculate accuracy of the symbolic models.