[Reviews · NeurIPS 2020]

Review 1

Summary and Contributions: This paper proposes a way to extract symbolic expressions for functions learnt by graph neural networks. Using these representations instead of black box DNN models encourages model discovery, interpretability, generalization and compactness. The authors show that they can distill analytical 'ground truth ' expressions and also generalize better.

Strengths: 1. It is a novel application of Graph neural networks for scientific tasks. The paper is a synergistic combination as it is the correct fit in terms of the data, gnn and task. 2. The paper’s interpretability results in the sense of how the `force' correlates with the learnt message vector is insightful.

Weaknesses: 1. The title seems misleading, the method from my understanding is a way to use GNNs for learning natural laws, or understand GNNs in terms of symbolic models, rather than extracting symbolic models from general deep learning models. If the latter is their main claim, it would be helpful if the authors show the utility of this model on more general tasks, other than natural laws. 2. The paper lacks a strong related work section, missing key details about models that have been used previously for relevant tasks. For example, the authors have not discussed how other papers like Neural Relational Inference or using Generative Models for their tasks. 3. The paper's main claim is interpretability and generalization, however while they reliably show that their model fits the ground truth equation, the authors do not show robustness or generalization on the simulation data at hand, which would be a better indicator of the claims of the model. 4. It would be helpful if the authors address the natural question how utilizing this model for a general deep learning model is more interpretable or generalizable especially in the cases of noise? 5. The paper needs to add more details in the main draft. Key details are in the appendix, the authors should include majority of Section A.1 in the main paper rather than the appendix. The simulation details should be explained more, are the authors generating time series datasets, the number of samples, length of the time series, how the trajectory is processed etc. While some details are covered in the appendix, these details are essential to understand the task and model.

Correctness: Yes

Clarity: The paper is mostly well written, however the details are scattered across the paper and it is hard to keep track. It would be helpful if the authors found a way to concisely explain the model more concretely together in one place. The specific case studies are only overviews without details, making it harder for the general reader to understand. For example, the symbolic model discussion in Section 2 should include more details of the input and output, corresponding dimensions for each node and layer. Additionally, key details are in the appendix, like section A.1. While the paper provides a decent overview of the task and the model, the main paper should include essential details as well.

Relation to Prior Work: The authors have not explained the previous work well. The authors should include a more detailed section on hamiltonian GNNs, their simulation case studies, etc.

Reproducibility: Yes

Additional Feedback: 0. The notations in the method section especially Section 2 need to be specified, even if it is easy to infer from context,. For example, L_v, v_i, v_j etc. need to be explained. Further, in the case studies sections, the descriptions are not clear, for example, the system should be explained mathematically from a n-body perspective, clearly denoting the particles as nodes at gnn equation level for atleast one case. 1. The authors should discuss the intuitions behind their specific model decisions, for example, as this is a model discovery task, why haven't the authors used generative model frameworks? 2. The input/output dimensionality for eureqa fitting should be explained in Section 3, for example, GNs have multiple layers, how does the proposed method fit equations for the edge/node functions at different layers and put them together? From the simulation dataset, the underlying model does not seem to need multiple layers for GNs. 3. The Hamiltonian Dynamics section is very hard to read, especially to a non-physics person, it would be helpful if the authors add a clear description of the input (like position and momentum ) and output for the HGN. 4. What is the intuition behind the sum of pairwise and self for the HGN? Have the authors compared to a model without this assumption? 5. Does the Bottleneck model perform worse simply because its a much smaller model than the other models with a large hidden size? 6. Line 170 states that "models are trained to predict acceleration given current state", do the authors not account for time dependence? Would adding a temporal loss improve their results, as this is an additional inductive bias? 7. Why have the authors used only the most significant component as input to Eureqa? Have they experimented with ways utilizing all the dimensions? 8. It would be helpful for general readers to know the technical challenges that the authors encountered in the proposed method. Typos: line 174 two tables.


Review 2

Summary and Contributions: In this paper, the authors proposed a general approach to distill symbolic representations of a learned deep model by introducing strong inductive biases. The method trained GNN and deployed symbolic regression to components of the learned model to extract explicit physical relations. The model was capable to capture some existing equations, including force laws and Hamiltonians, and to predict the concentration of dark matter from the mass distribution of nearby cosmic structures.

Strengths: The framework assembled several AI techniques and successfully applied in computer-aided theorem proving and discovery. The model not only recovered the injected closed-form physical laws for Newtonian and Hamiltonian examples, but also derived a new interpretable closed-form analytical expression that can be useful in astrophysics. However, I don’t have the expertise to evaluate the newly derived physical analytical expression.

Weaknesses: The work looks a bit weak theoretically, from perspective of AI research. I’m not sure if NeurIPS is the most suitable conference to publish this work.

Correctness: Yes.

Clarity: Basically yes.

Relation to Prior Work: Yes.

Reproducibility: Yes

Additional Feedback:


Review 3

Summary and Contributions: The paper presents a two-stage method for finding symbolic descriptions of physical data, which involves first fitting a graph network to the raw data (with regularization to encourage compactness) and afterwards running the symbolic regression package eureqa to find an analytic expression for the most "significant" message component.

Strengths: The test of generalization, in which the symbolic model induced by regression on graph network messages generalizes better than the graph network itself, is exciting and points to a promising research direction. The difficulty of the symbolic regression problem tackled in the dark matter example is also impressive; the majority of prior symbolic regression work focuses on test functions which are in the grammar of the regression solver (things like simple polynomials, logarithms, etc), or known physical laws (as in the first two case studies in this paper). I am not aware of prior work which uses neural nets combined with symbolic regression to try to find *new* physical laws and takes seriously the answers provided by the algorithm as serious candidates for study by physicists.

Weaknesses: Overall, the paper comes off as lacking some precision in its algorithmic description. At multiple points I would have liked a formal statement of, for example, an objective function to accompany sentences like: L211: "[we] select the fit that maximizes the fractional drop in mean absolute error (MAE) over the increase in complexity for the next best model." This objective is indeed included in the appendix (L594), but it would not take up much space at all to also include these in the main body and would help make the paper more precise. Additionally, the details of using an off-the-shelf symbolic regression package will likely be opaque to the general NeurIPS community. The precise objective is specified in the appendix, and the primitives and dataset size are given in the main body, but there are other considerations and parameters that might be of interest here. Did you set a time limit on the symbolic search? Which genetic algorithm is used? Can you control the "frontier size" as in search algorithms like BFS? The general outline of including (sometimes redundant, high-level) method descriptions for each of the three cases studies in the results section, as opposed to one complete description of the entire pipeline, makes it somewhat more difficult to piece together the full algorithm.

Correctness: It would be helpful to include a baseline (possibly from [34], as discussed in the "Relation to prior work" section) to better underscore the improvements of this method over prior work. I would not be surprised if prior methods fail in the dark matter case study, but it does seem like Newtonian and Hamiltonian dynamics are within scope of earlier algorithms.

Clarity: See concerns in "Weaknesses" and "Relation to prior work" sections.

Relation to Prior Work: I am not convinced this paper frames itself clearly with respect to prior work. It would be fair to expect some discussion of prior symbolic regression methods, in particular those combined with deep networks, but these are mostly lumped into this single sentence: L85: "There are other great alternatives and different approaches to eureqa, including [14, 7, 27, 1, 34]" without any further explanation of the differences between prior work and the proposed method. [34], for instance, also uses neural networks to "recursively break hard problems into simpler ones with fewer variables" for symbolic regression of 100 equations from the Feynman Lectures. This is superficially very similar-sounding to this paper's use of graph networks to break an intractable search problem into one of fitting $\phi^e_1$. [34] also includes an empirical comparison to eureqa, and seems like it would be a good candidate for a baseline for this paper, but at the very least slightly more discussion is warranted. Other works which go unmentioned include: 1. Peterson. Deep Symbolic Regression: Recovering mathematical expressions from data via risk-seeking policy gradients. 2020. 2. Kusner, Paige, Hernandez-Lobato. Grammar Variational Autoencoder. 2017. 3. Sahoo, Lambpert, Martius. Learning Equations for Extrapolation and Control. 2018.

Reproducibility: Yes

Additional Feedback: I applaud the authors for including a Google Colab link. This somewhat mitigates my concerns about precision and clarity in the writing (though it would still be better if the paper had a more thorough description as a stand-alone offering). Overall, I do lean positive on this paper, since the dark matter case study is very promising and might point to a more general principle relating to the generalization of deep nets versus symbolic expressions. However, there is currently little discussion of prior work attempting to solve the same problem, and such a discussion (or even empirical comparison) would make this paper both stronger and easier for readers to contextualize.


Review 4

Summary and Contributions: I have read the author response and reviewer discussion. The paper proposes a method to discover simple algebraic equations that describe physical systems. The method first fit a Graph Neural Network to observed data from the physical systems. It then use a commercial symbolic regression package to find a symbolic expression that fit the learned functions of the graph neural network. The proposed method is applied to 3 different domains, 1) newtonian physical systems, 2) hamiltonian physical systems and finally 3) a dark energy domain in which the true physical relationship is unknown. The paper shows that with sufficient regularization of the graph neural network representation the recovered representations are the expected physical forces up to a rotation. Given this representation the commercial symbolic regression package recovers several known physical laws. The paper uses the proposed method to discover a new equation that describe some property of dark matter, which is better than a hand crafted equation.

Strengths: The paper addresses an important question: how to discover simple algebraic equations which explain and predict a system. This could have far reaching implications, significantly advancing science in multiple fields, where closed form expressions of studies systems are not discovered. While far from perfect it makes good progress. The evaluation is done across a sufficient number of domains. The overall approach is described well and the paper is generally well written. Code is released along with the paper.

Weaknesses: Some sections of the paper are not very clear. Notable the section on Hamiltonian networks is hard to understand, and the description of the dark energy task doesn't introduce the problem sufficiently well. The paper should not assume the reader knows anything about dark energy cosmology. delta and rho on line 232 are undefined. The evaluation could be made more clear. The paper notes that "we recover many analytical expressions". How many is many? It then lists a few samples along with expected analytical formula, although it's not immediately clear whether these are correct. For instance the expected formula for "1/r2, 3D, Bottleneck" include delta z, which is not present in the discovered expression. I'd recommend revising this presentation, and show a table with all the experiments and a simple boolean whether the method found the expected formula or not. Similarly it's hard to compare numbers in table 1, since MSE is not a very easy metric to understand intuitively. How much worse is 1e-5 than 1e-6 really? Maybe using correlation or R^2 or another metric would make it more clear. I did not understand table 2. What is "formula" and what is "summed component"? I could not parse the sentence starting on line 292. "We introduced..." please revise. 246: missing "to" 194: spurious "table" 185: "We speculate this is because the low-dimensionality of the messages makes it harder to solve the optimization process via gradient descent." Why? Lottery ticket hypothesis? Please provide an argument for this speculation. 177-178: revise sentence. 135: spurious momentum Broader impact section doesn't really consider the broader societal impacts. It re-iterates the papers main points and describes possible future work. To be fair I don't really know what the broader impact of this paper is either.

Correctness: Yes. The metrics and presentation of results can be improved though.

Clarity: Large parts of the paper is well written. A few sections needs work. The main point definitely comes across.

Relation to Prior Work: Prior work is not discussed in detail. There's no comparison to previous work.

Reproducibility: Yes

Additional Feedback: The paper finds that the simple algebraic equations discovered generalize better to new data than the underlying graph networks. It would be interesting to see if one could find the simple equations directly by parameterizing the graph neural networks in terms of the terms which the symbolic regression engine uses, e.g. plus, minus, power, etc.

[Author Response · NeurIPS 2020]

We thank the reviewers for giving positive and insightful evaluations of our paper. We will adjust the camera-ready
version to improve clarity of explanations and address all other comments. Specific responses are given below.

**Related work.** Thank you for introducing us to two new symbolic regression (SR) papers: Peterson (2020), and Sahoo
et al. (2018). We will discuss these in our paper. Note that our work is slightly orthogonal to these approaches: SR
algorithms like these and Eureqa, dCGP, and [34] are techniques to search an equation space with few input variables in
the raw symbolic regression setting. Our paper gives a way of extending any of these to high-dimensional problems, by
factorizing the problem into low-dimensional sub-problems corresponding to a neural network's structure. Consider
attempting to fit the equivalent of both the edge model and node model, simultaneously, with any SR approach. In a
typical setup, the number of possible functional forms for these under a complexity upper bound of 10 tokens, and
only one latent variable, is approximately $10^9$ equations each, but explored simultaneously, there are $(10^9)^2 = 10^{18}$
possibilities. However, by using our strategy to learn these models with a neural network first, and fit symbolic regression
to the edge and node model independently, we go from $10^{18}$ to $2 \times 10^9$ considered equations. This speedup grows for
multiple latents. Future work could replace Eureqa inside our framework with more sophisticated SR backends.

**Comparison of SR packages.** We will discuss a comparison of these SR packages, hyperparameters (fixed in Eureqa),
and why we chose Eureqa—we realize now that many at NeurIPS would find this useful. Furthermore, we will
demonstrate a comparison of our approach against a pure SR on one of our case studies. We have previously performed
this experiment and found that Eureqa performed very poorly without our framework.

**Udrescu & Tegmark (2019) [34].** This Eureqa alternative is optimized for rediscovering existing equations by, e.g.,
requiring as input the equation's constants, which allows for dimensional analysis (a trick to recall equations in
physics). This approach does not seem applicable for discovering new equations so we chose Eureqa. Their description
"recursively break hard problems into simpler ones with fewer variables" refers to a common search strategy in SR to
determine feature importance (related to feature selection in decision tree learning), dissimilar to our framework.

**Other architectures.** Our procedure is not restricted to GNNs. However, the inductive bias of the architecture
determines the structure of the recovered equation. GNNs with sum-pool give the form $y_i = f(x_i, \sum_j g(x_i, x_j))$. We
focused on GNNs since this equation matches our problems, yet one can apply our framework to alternate inductive
biases. We will add a discussion on additional example architectures.

**Other domains.** Regarding tests for alternate domains, symbolic priors might work best for problems where we know
simple analytic equations already predict accurately: physics, chemistry, engineering, etc., whereas, e.g., biology and
behavioral sciences lack strong analytic models. Perhaps there exists a deep unknown reason for this, or maybe one
could use our framework to discover new equations in these domains. Regardless, we hope that our paper will stimulate
future research to apply our framework to new architectures and datasets.

**Lottery ticket hypothesis.** We thank the reviewers for making this connection to the bottleneck model results. We will
incorporate this in the discussion.

**Generalization on simulation data.** We did not test generalization performance on our simple n-body datasets because
our framework recovered the ground truth equation. Thus, since Newton's law generalizes to any number of bodies,
this equation will give a loss of zero on any dataset with the same force law. However, in the dark matter experiment,
we also demonstrate better generalization (Section 4.3 - Symbolic generalization) than the trained neural network,
despite this simulation being entirely unlike a simple analytic equation. This simulation integrates the complex and
noisy (chaotic) dynamics of dark matter for a million CPU hours. Yet the analytic equation obtained via our framework,
describing a pattern in the output dark matter dataset, generalizes significantly better (error=0.0892) than the same
neural network it was extracted from (error=0.142). The question: "Where and why does a symbolic prior improve
generalization?" presents a very intriguing problem and motivates future research.

**Metrics.** We will update Table 1 to use $R^2$ as a metric instead of mean square error to make results more intuitive.

**Quantifying SR.** As suggested, we will create a table showing success/failure for each force law SR reconstruction.

**Table 2.** "Formula" represents the recovered analytic expression to predict overdensity. These formulas contain a
variable "$e_i$", which represents a sum of some scalar function over nearby dark matter, given in the adjacent column.

**Generative models.** We have so far not experimented integrating generative models in our framework, but this sounds
like a very interesting approach, and we look forward to exploring this direction in future work.

**Broader impact.** We are also unaware of potential misuses, but will discuss any suggestions.

**Global models.** We studied GNNs with an edge model and node model (Newtonian, dark matter), a global model
(Hamiltonian), but not all three models together. One could approach such a GNN with the same technique: sparsify
latents, apply SR to each model, compose.

[Meta-Review · NeurIPS 2020]

Paper presents an exciting area of research. All reviewers agree that the paper makes novel contributions. The one weak point of the current submission is that this work is not properly contextualized with prior work. Further, as authors said in their rebuttal -- it would be good to see comparisons with other SR packages and SR only baseline.